# Ester Production Using the Lipid Composition of Coffee Ground Oil (*Coffea arabica*): A Theoretical Study of Eversa® Transform 2.0 Lipase as an Enzymatic Biocatalyst

Millena Mara Rabelo Nobre [1], Ananias Freire da Silva [1], Amanda Maria Menezes [1],
Francisco Lennon Barbosa da Silva [1], Iesa Matos Lima [1], Regilany Paulo Colares [2],
Maria Cristiane Martins de Souza [1], Emmanuel Silva Marinho [3], Rafael Leandro Fernandes Melo [4],
José Cleiton Sousa dos Santos [1,*] and Aluísio Marques da Fonseca [2,*]

1 Instituto de Engenharia e Desenvolvimento Sustentável, Universidade da Integração Internacional da Lusofonia Afro-Brasileira–UNILAB, Redenção 62790-970, CE, Brazil; millena_mara@hotmail.com (M.M.R.N.); ananiasfreire21@gmail.com (A.F.d.S.); meneses.amandamaria@gmail.com (A.M.M.); lennonsilva1717@gmail.com (F.L.B.d.S.); iesamattos@gmail.com (I.M.L.); mariacristiane@unilab.edu.br (M.C.M.d.S.)

2 Instituto de Ciências Exatas e da Natureza, Universidade da Integração Internacional da Lusofonia Afro-Brasileira–UNILAB, Redenção 62790-970, CE, Brazil; regilany@unilab.edu.br

3 Faculdade de Filosofia Dom Aureliano Matos, Universidade Estadual do Ceará, Limoeiro do Norte 62930-000, CE, Brazil; emmanuel.marinho@uece.br

4 Departamento de Engenharia Metalúrgica e de Materiais, Universidade Federal do Ceará–UFC, Campus do Pici, Fortaleza 60714-903, CE, Brazil; rafael.melo@ifce.edu.br

* Correspondence: jcs@unilab.edu.br (J.C.S.d.S.); aluisiomf@unilab.edu.br (A.M.d.F.)

**Abstract:** The scientific community recognizes coffee grounds (*Coffea arabica*) as an important biological residue, which led to using the Eversa® Transform 2.0 lipase as an in silico enzymatic catalyst for coffee grounds' free fatty acids (FFA). Molecular modeling studies, including molecular docking, were performed, which revealed the structures of the lipase and showed the primary interactions between the ligands and the amino acid residues in the active site of the enzyme. Of the ligands tested, 6,9-methyl octadienoate had the best free energy of −6.1 kcal/mol, while methyl octadecenoate and methyl eicosanoate had energies of −5.7 kcal/mol. Molecular dynamics confirmed the stability of the bonds with low Root Mean Square Deviation (RMSD) values. The MMGBSA study showed that methyl octadecenoate had the best free energy estimate, and CASTp identified key active sites for potential enzyme immobilization in experimental studies. Overall, this study provides efficient and promising results for future experimental investigations, showing a classification of oils present in coffee grounds and their binding affinity with Eversa.

**Keywords:** coffee grounds; Eversa® Transform 2.0 lipase; molecular docking

## 1. Introduction

Coffee has a global consumption of more than 10 million kilos per year, being considered one of the most popular hot drinks in the world [1]. Soluble coffee production has a 1:2 ratio of coffee grounds generation and wet coffee grounds. That is, the large consumption of this food generates excessive biological waste with high energy potential [2,3]. These residues can be separated and processed, generating new products with the possibility of generating added value. Some bioactives that can be obtained from coffee bagasse are lignin, cellulose, and hemicellulose, which are highly viable for producing biomass and obtaining biofuels. In addition to this generation of value, these bioactives do not compete with the coffee food chain, which is another advantage related to their study [4,5].

The rapid growth of non-renewable fuel exploitation, such as oil, is a significant concern. If economic growth continues at its current rate, energy demand will increase

tenfold by 2050 [6]. Consequently, using renewable fuels like bioethanol and biodiesel is highly encouraged, and the potential of coffee grounds for biofuel production should not be overlooked [7].

For free fatty acids (FFA) obtained from coffee bagasse to be used as biofuels, the FFA need to undergo a catalysis process (being esterified or transesterified) to become viable for use [8,9]. This catalysis can be obtained chemically or enzymatically [10,11]. However, chemical catalysts are limited by their longer reaction time and corrosive nature, making enzymatic catalysts a more favorable option [12,13]. Although enzymatic catalysts still have a high cost, the scientific community has worked to reduce this cost by various means, such as reactor design, reaction and enzymatic engineering, use of alternative cofactors, and use of low-cost enzymes, among others. One of the ways that has been most studied for this cost reduction is the process of immobilizing enzymes in non-soluble supports to form a biocatalyst that can be recovered and reused, making the process more efficient and sustainable [14,15].

Numerous studies have been conducted in the field of enzymatic catalysts to reduce the cost of the process, both by immobilizing the enzymes and obtaining more accessible enzymes. In this regard, Eversa® Transform 2.0 lipase (EVS) is a more cost-effective alternative as it is produced by *Thermomyces lanuginosus* and synthesized from the genetically modified strain of *Aspergillus oryzae* [16,17]. Its enzymatic component, carboxylic acid ester hydrolysis (EC 3.1.1.3), has emerged as a promising candidate for producing biolubricants and biofuels [18,19].

A factor to consider in reducing the cost of enzymatic catalysts for fatty acids is achieving optimal compression of the process and stability in binding the enzyme's active site with the fatty acids being used [20,21]. Achieving optimal compression involves examining systems at the atomic and molecular levels, which requires the study of computational chemistry and bioinformatics [22]. Understanding the affinity of the compounds, their enantioselectivity, or possible catalytic deactivations provides valuable data for designing an effective FFA catalysis strategy during experimental processes [23].

Techniques such as docking and molecular dynamics (MD) can be used to analyze the catalytic efficiency of a given lipase in various esterification and transesterification processes, providing an understanding of the interaction between the enzyme, substrate, and solvent without the need for expensive materials [24,25]. By gaining this understanding and using more affordable enzymes along with coffee grounds and biomass with biological potential that would otherwise be discarded, the production of biofuels has the potential to become both practical and cost-effective [26,27].

This study aims to evaluate the potential of Eversa® Transform 2.0 lipase to catalyze the seven fatty substances extracted from coffee grounds, including hexadecanoic acid, methyl hexadecanoate, methyl octadecanoate, methyl docosanoate, methyl eicosanoate, methyl octadienoate, and methyl octadecenoate, through molecular docking and dynamic simulations. Molecular docking was used to calculate and identify the ligands' conformational positions in the lipase's catalytic site, as well as the nature and amount of intermolecular interactions. Molecular dynamics was used to evaluate the stability of the enzyme–substrate complexes. The in silico results could lead to the better and more efficient production of biofuels from coffee grounds.

## 2. Materials and Methods

### 2.1. Homology Modeling

First, four-step comparative modeling of the Eversa® Transform 2.0 (São Paulo, SP, Brazil) lipase protein [20] was performed.

### 2.1.1. Identification and Selection of Protein-Fold

In order to identify a related protein to the amino acid sequence of EVS (CAS number 9001-62-1 from Sigma-Aldrich (São Paulo, SP, Brazil)), we used the BLAST program (Basic Local Alignment Search Tool (Public domain, Bethesda, MD, USA)) [28] and its PDB

database for comparative analysis. This led us to identify a hydrolase enzyme, expressed by the *Escherichia coli-Pichia pastoris* shuttle, from the organism *Aspergillus oryzae* with the code 5XK2 in the Protein Data Bank as the target protein.

### 2.1.2. Alignment of Target and Mold Sequences

Alignment between the sequences was performed using the Modeller software (ver 10.4, San Francisco, CA, USA) [29].

### 2.1.3. Model Construction and Optimization

The model was constructed using the Modeller 10.4 *software* (ver 10.4, San Francisco, CA, USA) [29], resulting in a new protein named EVS that was evaluated for function, target, and stereochemical parameters [30].

### 2.1.4. Protein Validation

The model was validated at the stereochemical, conformational, and energetic levels. The three-dimensional structure of the generated model was evaluated for possible stereochemical quality using the PROCHECK software (ver 2023, Hinxton, Cambridgeshire, CB10 1SD, UK), which included the validation of the model by the Ramachandran plot [31].

### 2.2. Protein Preparation

The protein generated by EVS homology was subjected to correcting charges and adding hydrogen atoms using the AutoDock Tools software (ver 1.5.7, California, CA, USA) [29].

### 2.3. Obtaining the Ligand

The lipid composition structures of *Coffea arabica* oils (Figure 1) were generated using ChemDraw 3D software (ver 18.1, New Jersey, NJ, USA) and then minimized with an RMS gradient of 0.0001 using an MM2 force field [32]. The structures were further optimized using Avogadro® software (ver 1.2, Pittsburgh, PA, USA) [33], using the Merk molecular force field (MMFF94) with a convergence limit of $10 \times 10^{-7}$ and 500 interaction cycles before being converted to PDBQT format [34].

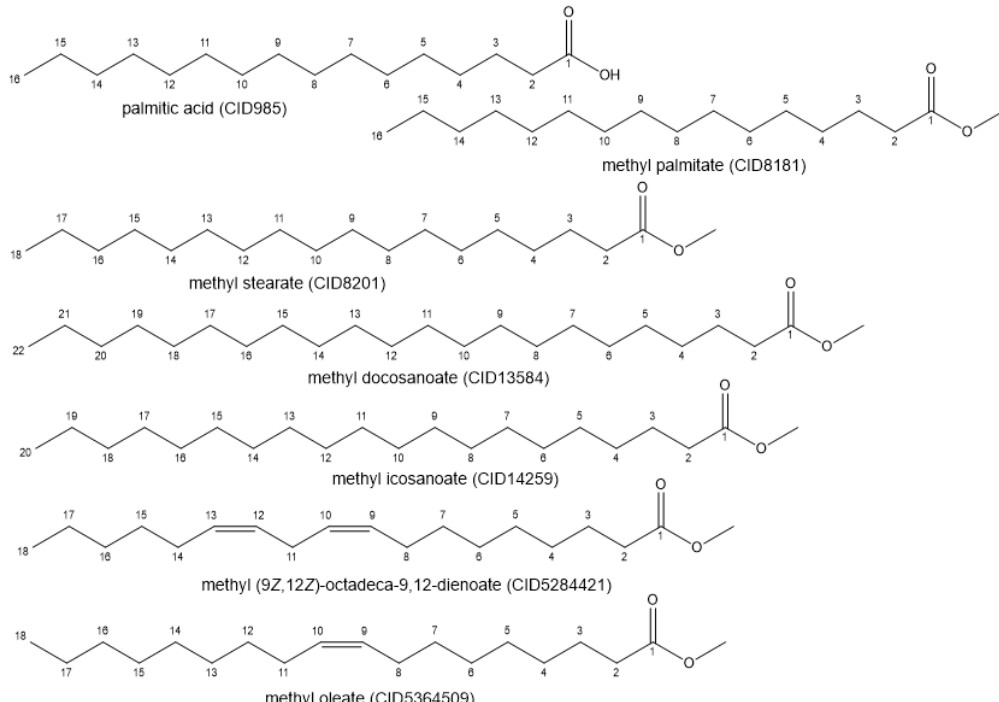

**Figure 1.** 2D Structure of the lipid composition of *Coffea arabica*.

*2.4. Molecular Docking and Visualization of Calculations*

The process of molecular coupling simulation was performed using the AutoDock Vina code (ver 1.2, California, CA, USA) [35] with rigid proteins and flexible ligands. A lattice configuration with enzyme active site parameters was set up for both calculations [36,37]. The software was used to evaluate the energy profiles of the ligand-receptor interactions. The PyMol software (ver 2.5, California, CA, USA) was used to visualize the anchored positions [38].

AutoDock Vina was used to calculate the stability of the enzyme substrate using statistical parameters such as RMSD (Root Mean Square Deviation) with a cut-off value of 2.0 Å [37] and affinity energy, with a cut-off value of −6.0 kcal/mol [38]. The intensity of the hydrogen bonds (H-Bond) was assessed by the distance between the donor and acceptor atoms, classified as Strong bonds (2.5–3.1 Å), Average bonds (3.1–3.55 Å), and Weak bonds (greater than 3.55 Å) [39]. The affinity energy was also used to assess the stability of the formed complexes.

To find immobilization binding sites, CASTp 3.0 was used to identify and measure accessible surface pockets [40]. Several criteria had to be met for the selected immobilization to be accepted: the site had to be distant from the active catalytic center of the enzyme so as not to affect the catalytic activity; the size of the site/pocket should be large enough to accommodate the selected affinity binding; and finally, the surface properties of the site region had to be significantly different from those of the active site.

*2.5. Molecular Dynamic*

Molecular dynamics (MD) simulations were carried out using the NAMD program [41–43]. The optimal conformations obtained from molecular docking were solvated in water using the TIP3P model, with the CHARMM36 force field, and ions were added to neutralize the overall system charge. The system was then energy-minimized using the Steepest Descent method and subjected to NVT and NPT equilibration under Langevin conditions [44]. Production simulations were conducted for a duration of 100 ns. The quality of the structures obtained in MD simulations was evaluated using the following parameters with NAMD (Nanoscale Molecular Dynamics):

- Potential energy (kcal/mol) [45];
- Protein–ligand interaction energy (kcal/mol);
- The root mean square deviation of the atomic positions of proteins, binders, and the distances between them (RMSD, Å), and the root mean square deviation of the atomic positions of proteins, ligands, and the distances between them (RMSD, Å);
- Hydrogen bonds were evaluated using Visual Molecular Dynamics (VMD) [46];
- The mean square fluctuation of the minimum distances between proteins and ligands was observed in MD (RMSF, Å) [47]. The plots were generated using the Qtrace program.
- In this study, MD simulations were used to evaluate the stability of a viral protease enzyme with various ligands containing different amounts of α-helix and β-sheets [48]. The long-range interactions were calculated using the SPME method and a Langevin thermal bath at 310 K. The conformational changes of the protein during the MD simulations were described using root mean square deviations (RMSD).

MM/GBSA Calculations

Based on the MD log file of the NAMD software (ver 2.14, Illinois, USA) [43], the MM/GBSA was calculated by MolAICal (ver 1.3, Gansu, China) [49] and estimated using Equations (1)–(3).

$$\Delta G_{bind} = \Delta H - T\Delta S \approx \Delta E_{MM} + \Delta G_{sol} - T\Delta S \tag{1}$$

$$\Delta E_{MM} = \Delta E_{internal} + \Delta E_{ele} + \Delta E_{vdw} \tag{2}$$

$$\Delta G_{sol} = \Delta G_{GB} + \Delta G_{SA} \tag{3}$$

where $\Delta EMM$, $\Delta G_{sol}$, and $T\Delta S$ represent the gas phase MM energy, the solvation-free energy (sum of the polar contribution $\Delta G_{GB}$ and the nonpolar contribution $\Delta G_{SA}$), and the conformational entropy, respectively [48]. $\Delta E_{MM}$ includes the van der Waals energy $\Delta E_{vdw}$, the electrostatic energy $\Delta E_{ele}$, and the $\Delta E_{internal}$ bond, angle, and dihedral energies. If there are no bond-induced structural changes in the process of MD simulations, the entropy calculation can be omitted [49].

## 3. Results and Discussion

### 3.1. Immobilization Locations

To identify binding pockets suitable for immobilization, the CASTp tool was used to locate accessible surface pockets [40,50]. Seven pockets located away from the catalytic triad—equivalent to the enzyme's active site—were selected for immobilization, as shown in Figure 2. This approach helps to prevent interference with the biocatalytic process. This strategy minimizes interference with the enzyme's catalytic activity compared to other immobilization methods. Moreover, immobilizing the enzyme at sites distant from the active site may enhance its stability, extend its lifespan, and reduce the cost of biocatalyst replacement.

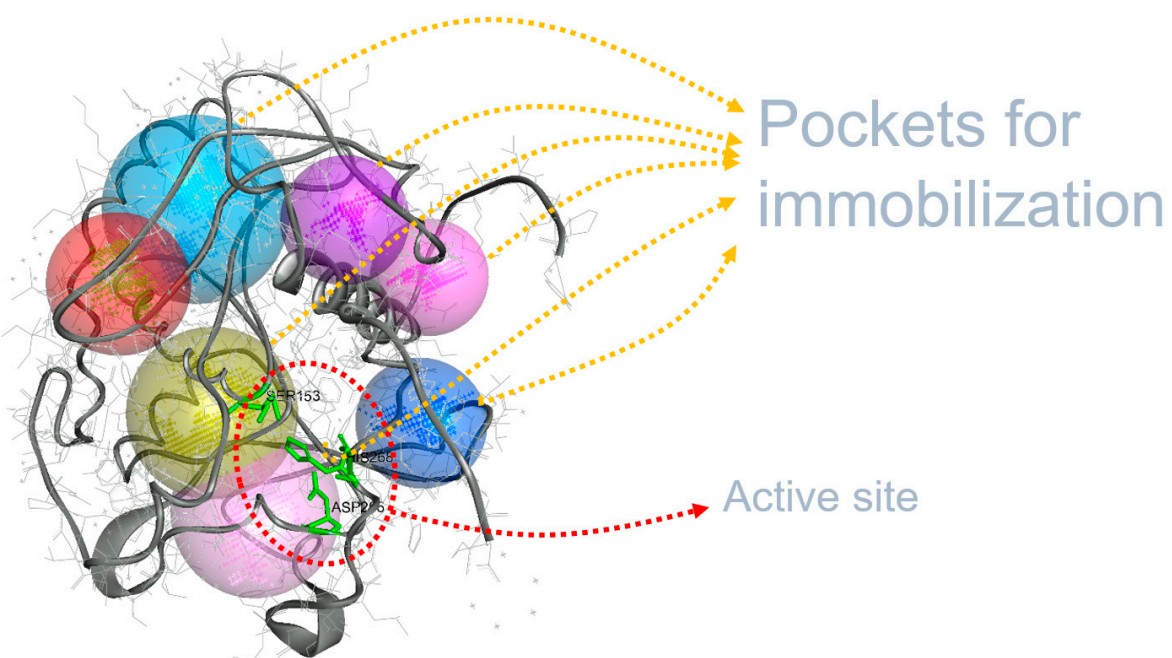

**Figure 2.** Detection of seven cavities using CASTp.

### 3.2. Protein Modeling

The protein structure quality was assessed using the Ramachandran plot (Figure 3), which displays the distribution of amino acid residues in different regions of the protein structure based on their backbone torsion angles. Most residues (91.5%) were located in the favorable regions (red region), indicating a high-quality protein structure. A small percentage of residues (6.5%) were found in additionally allowed regions (a, b, l, p regions, yellow), while an even smaller percentage (1.6%) were in the generously allowed parts (~a, ~b, ~l, ~p regions, light yellow). Only a minority of residues (0.4%) were located in the unfavorable regions (empty region), which could be attributed to using templates for the protein structure prediction and some residues at the ends of the protein. These results support the reliability of the protein model obtained.

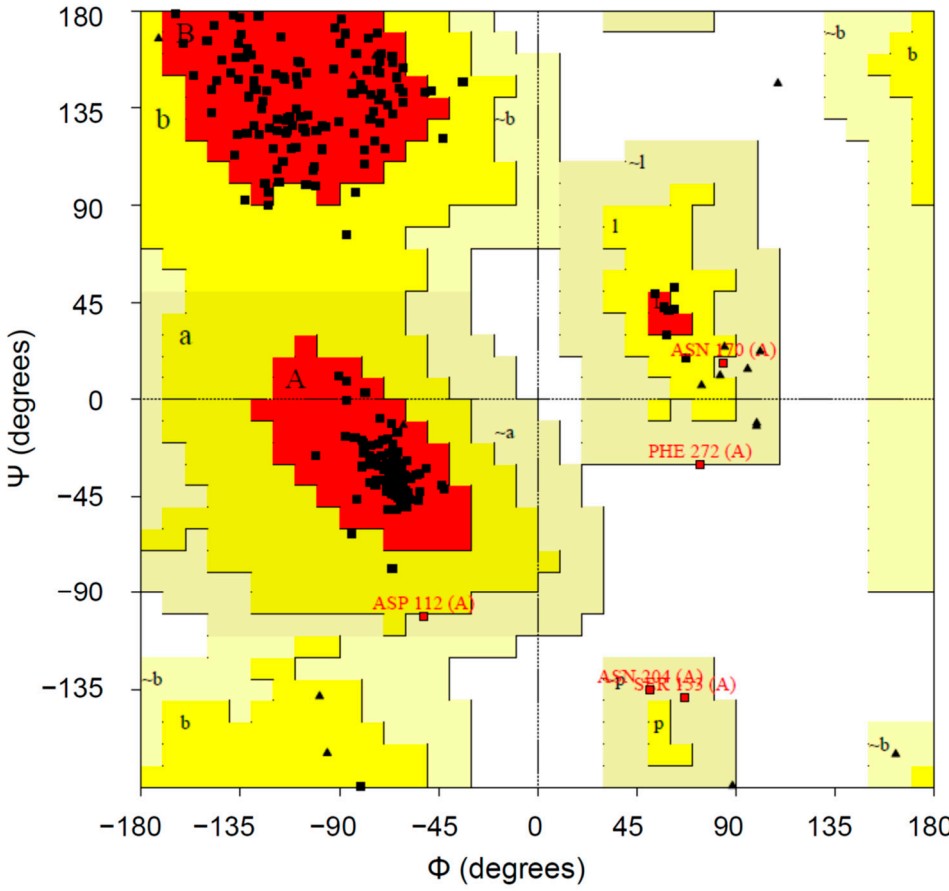

**Figure 3.** Ramachandran plot of Eversa modeled.

In addition, performing a structural alignment of the predicted protein with other known proteins enabled the identification of both conserved and variable regions in the protein's primary sequence. Analyzing structurally equivalent residues in different proteins made it possible to determine which protein regions are crucial for their structure and function. This approach was utilized to identify essential residues in the lipase protein under investigation [51].

### 3.3. Interaction between Substrate and Lipase

Molecular pairing studies were performed to validate the approaches used to explain the observed results for EVS. Consistent with the van der Waals forces reported in the literature, hydrogen bonds were favorable with binding affinities indicated by molecular coupling studies [52]. Therefore, for immobilization purposes, Eversa® Transform 2.0 lipase was structurally studied by molecular modeling with a lipase binding survey using AutoDock Vina to predict its affinity, orientation, and environmental surfaces [53].

The EVS catalytic site is a triad represented by residues Ser 153, His 268, and Asp 206 [54], of which the serine residue acts as a nucleophile on the substrate carbonyl group for esterification bioreactions only within the substrate pocket [55]. Only substrates of suitable molecular forms can occupy these subsites and undergo catalysis, such as the carboxylic acids and esters on the coffee grounds oil composition.

The estimated binding affinity between the anchored composition oil and the enzyme ranged from −5.1 kcal/mol to −6.1 kcal/mol (Table 1). This lower binding energy indicates that the substrate and lipase combination was more stable and suitable for esterification. Figure 4 shows the simulation results in 2D.

**Table 1.** Oil composition and molecular docking results.

| Sample | Energy (kcal/mol) |
|---|---|
| CID985 hexadecanoic acid | −5.6 |
| CID8181 methyl hexadecanoate | −5.4 |
| CID8201 methyl octadecanoate | −5.6 |
| CID13584 methyl docosanoate | −5.4 |
| CID14259 methyl eicosanoate | −5.7 |
| CID5284421 6,9-methyl octadienoate | −6.1 |
| CID5364509 methyl octadecenoate | −5.7 |

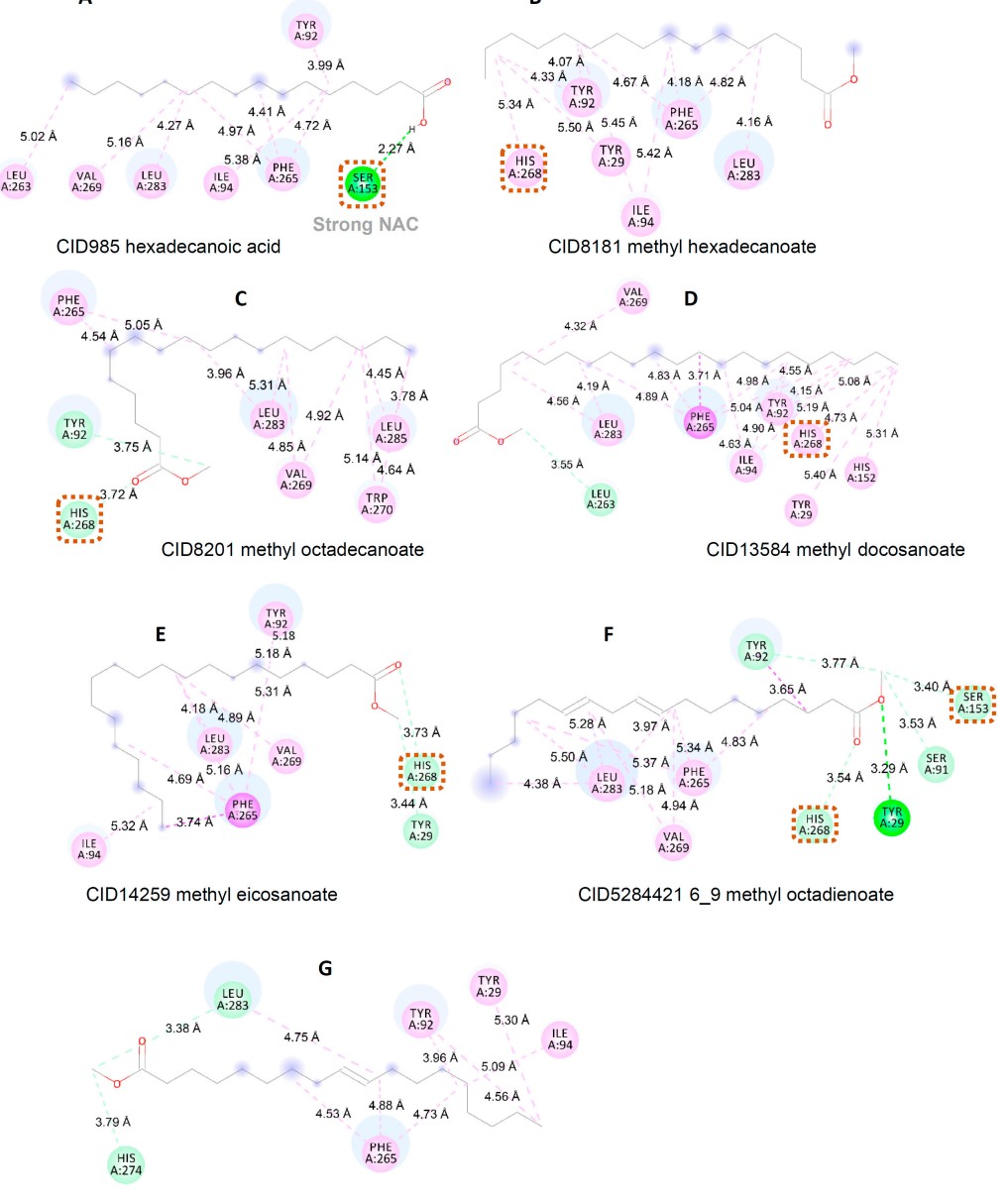

**Figure 4.** Substrate in 2D: (**A**) hexadecanoic acid; (**B**) methyl hexadecanoate; (**C**) methyl octade-canoate; (**D**) methyl docosanoate; (**E**) methyl eicosanoate; (**F**) 6,9-methyl octadienoate; (**G**) methyl octadecenoate.

The molecular docking study revealed that all other derivatives interacted with at least one of the catalytic triad residues except for the methyl compound octadecenoate.

Specifically, they were close to the carboxylic acid region of Ser 153 and His 268, which, according to the literature, slightly enhances the esterification reaction's ester formation [56]. The content framed in red is the amino acid residues that are part of the catalytic triad. For this reason, they were stated prominently (Figure 4).

The Near Attack conformations (NACs) refer to conformations consistent with the attack of the catalytic site on the electrophilic carbon of the acyl group [57]. Usually, in a NAC, the distance between the oxygen of the Ser 153 residue (EVS) and the carbonyl carbon is about 3 Å, and the same atoms with the carbonyl oxygen molecule tend to form an angle of approximately 60°, but up to 90° [58]. Therefore, the hexanoic acid compound exhibited a strong NAC.

Therefore, to delve deeper, this article examines the interactions between various chemical compounds and a particular enzyme. The immobilization of the enzyme is crucial in enhancing its stability and effectiveness for specific applications. The catalytic triad is a fundamental enzyme component that enables its biochemical function.

The results suggest that the catalytic triad will remain active after immobilization, and some oil compositions interact better with the enzyme than others. Specifically, 6,9-methyl octadienoate and methyl octadecenoate showed binding affinities of −6.1 kcal/mol and −5.7 kcal/mol, respectively. In the case of 6,9-methyl octadienoate, hydrogen bonding at Tyr 29, as well as conventional hydrogen–carbon interactions at Tyr 92, Ser 91, and the catalytic triad residues Ser 153 and His 268 were observed, indicating possible esterification reactions. Hydrophobic interactions were also noted at Phe 265, Val 269, and Leu 283. For methyl octadecenoate, two polar hydrogen–carbon interactions with Leu 283 and His 274, and apolar interactions with Tyr 29, Ile 94, Tyr 92, and Phe 265, were observed, but no interactions were seen with the catalytic triad. These interactions stabilize the enzyme and enable it to function, including hydrogen bonding, hydrogen–carbon interactions, and hydrophobic interactions, as depicted in Figure 5 and Table 2.

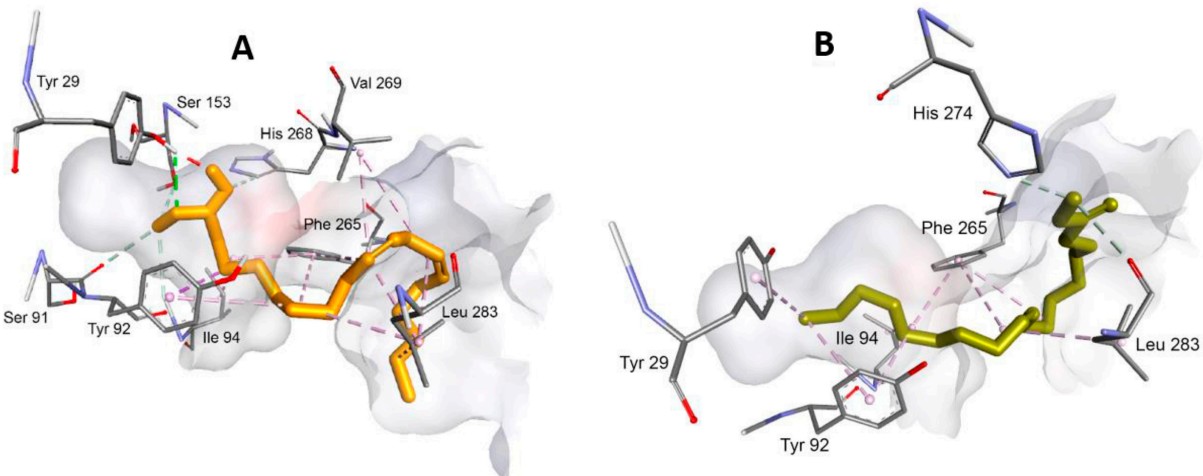

**Figure 5.** Interactions between 6,9-methyl octadienoate (**A**) and methyl octadecenoate (**B**) with Eversa amino acid residues.

**Table 2.** Interactions between amino acid residues and the lipid composition of coffee grounds.

| Sample | Residue | | | | | | | | | | | | |
|---|---|---|---|---|---|---|---|---|---|---|---|---|---|
| | Tyr 29 | Tyr 92 | Ile 94 | His 152 | Ser 153 | Leu 263 | Phe 265 | His 268 | Val 269 | Trp 270 | His 274 | Leu 283 | Leu 285 |
| CID985 | | 3.99 (HI) | 5.38 (HI) | | 2.27 (HB) | 5.02 (HI) | 4.41 (HI) 4.72 (HI) 4.97 (HI) | | 5.16 (HI) | | | 4.27 (HI) | |
| CID8181 | 5.50 (HI) | 4.33 (HI) 4.07 (HI) | 5.42 (HI) 5.45 (HI) | | | | 4.82 (HI) 4.18 (HI) 4.67 (HI) | 5.34 (HI) | | | | 4.16 (HI) | |
| CID8201 | | 3.75(CH) | | | | | 4.54 (HI) 5.05 (HI) 3.71 (PA) | 3.72 (CH) | 4.85 (HI) 4.92 (HI) | 4.64(HI) 5.14(HI) | | 3.96 (HI) 5.31 (HI) | 3.78 4.45 |
| CID13584 | 5.40 (HI) | 4.15 (HI) 4.55 (HI) 5.08 (HI) | 4.63 (HI) 4.55 (HI) 5.19 (HI) | 5.31(HI) | | 3.55(CH) | 4.83 (HI) 4.89 (HI) 4.98 (HI) 5.04 (HI) | 4.73 (HI) | 4.32 (HI) | | | 4.19 (HI) 4.56 (HI) | |
| CID14259 | 3.44 | | 5.32 | | | | 3.74 4.96 5.16 5.31 | 3.73 | 4.89 | | | | |
| CID5284421 | 3.29 | 3.65 | | | | | 4.83 5.34 5.18 | 3.54 | 4.94 5.18 | | | 3.97 4.38 5.28 5.50 | |
| CID5364509 | 5.30 | 3.96 | 5.09 | | | | 4.53 4.73 4.88 | | | | 3.79 | 4.75 | |

### 3.4. Molecular Dynamics

A model of a thermodynamic system consisting of a solute and a solvent can be constructed using a protein–ligand–solvent–ion complex. This complex involves various intermolecular forces and heat exchange between the molecules and ions. According to the laws of thermodynamics, the interactions between these molecules and the heat transfer process are influenced by various energy changes, as previously reported in the literature [59].

To gain a deeper insight into the behavior of protein–ligand complexes, molecular dynamics simulations were performed using NAMD [43]. These simulations aimed to evaluate potential global conformational changes and protein stability after each conformation and to obtain information on the mechanism of interaction between the complexes at the molecular level. Previous studies by Bylehn et al. (2021) and Du et al. (2016) have also used this technique to gain valuable insights into the structure and function of protein–ligand complexes [59,60].

The simulations' outcomes can have significant implications for various scientific and industrial fields, including drug development, biotechnology, and materials science. They can assist in developing drugs and therapeutics and optimizing chemical processes to enhance reaction efficiency, making them valuable tools for research and development [61].

### 3.4.1. RMSD Analysis

Soon after the molecular docking, the coffee bean oil composition was selected because it had the best binding energies to perform the molecular dynamics study according to the catalytic site of Eversa (Figure 6).

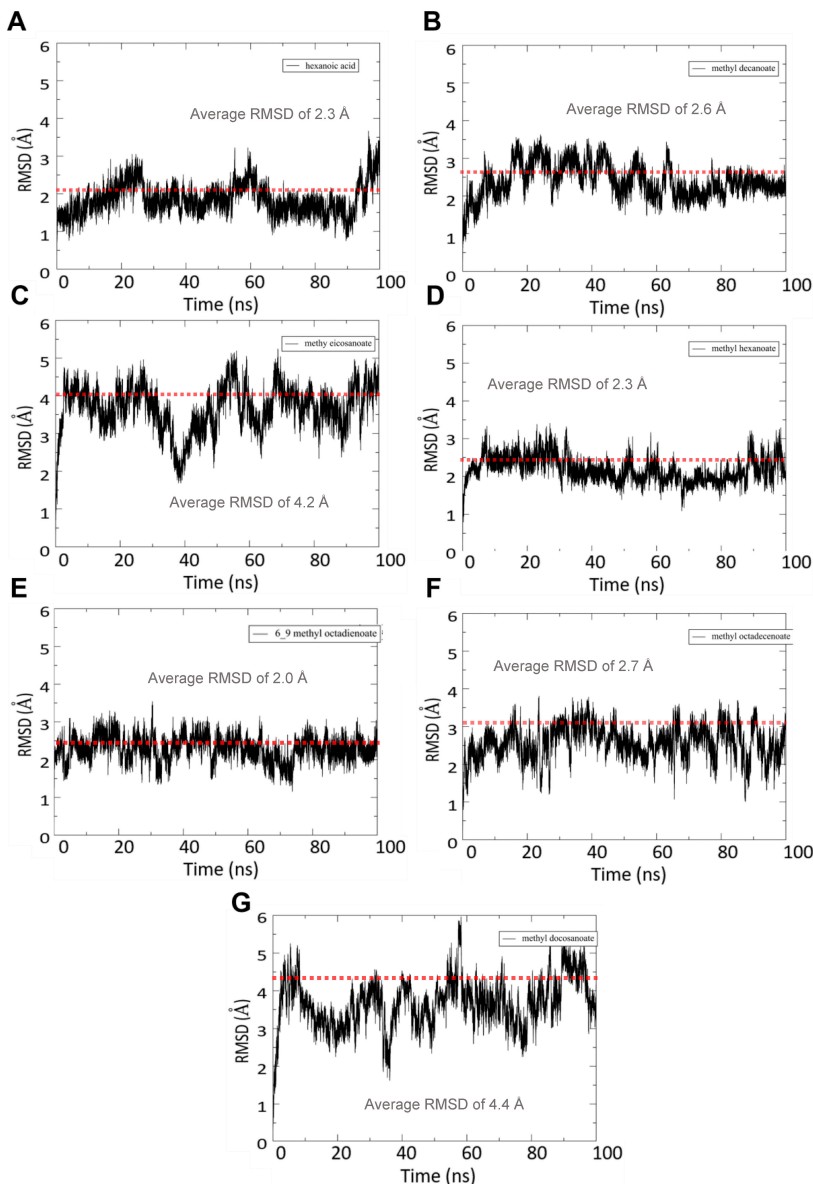

**Figure 6.** Root Mean Square Deviation (RMSD), concerning the initial confirmation of the ligand–enzyme complex versus the simulation time (ns) in the production simulations step of the MD with the ligand–enzyme complex versus the simulation time (ns) in the production simulations step of the MD with coffee ground oil composition/Eversa lipase. (**A**) hexadecanoic acid; (**B**) methyl decanoate; (**C**) methyl eicosanoate; (**D**) methyl hexadecanoate; (**E**) 6,9-methyl octadienoate; (**F**) methyl octadecanoate; (**G**) methyl docosanoate.

The simulations showed that Eversa maintained an average RMSD value of around 3.1 Å throughout the 100 ns production stages, while 6,9-methyl octadienoate showed a stable average RMSD value of 2.0 Å throughout the simulation. Hexanoic acid and methyl hexadecanoate showed excellent stability with an average RMSD value below 2.0 Å. However, some acids showed stable values with RMSD above 2.0 Å, consistent with previous studies such as Cavallari et al. (2006) [59]. The dotted red line signifies the average RMSD over the entire trajectory.

The conformational changes of the protein observed during MD simulations were characterized using the mean square deviations (RMSD) formula, as shown in Equation (4). The equation uses r*i*(*t*) and r*i*(0) to represent the coordinates of the *i*-th atom at times *t* and 0, respectively, while *N* denotes the number of atoms in the region of interest.

$$\text{RMSD} = \left[ \frac{1}{N} \sum_{i=1}^{N} \left[ \mathbf{r}_i(t) - \mathbf{r}_i(0)^2 \right] \right]^{1/2} \tag{4}$$

3.4.2. RMSF Analysis

Based on the RMSD analysis, the stability of the protein–ligand complexes was confirmed. However, to better understand the conformational changes observed during the molecular dynamics (MD) simulations, the atomic positions' root-mean-square fluctuations (RMSF) were calculated using Equation (5). RMSF values were determined by subtracting the average position of each atom $r_i$ from its position at each time step *j*, and then calculating the quadratic deviations. The total simulation time I was expressed as the total number of time steps collected, and this calculation provided detailed information on the protein dynamics during the simulations, including the impact of the interactions with the formed complexes [62].

$$\text{RMSF}_i = \left[ \frac{1}{\Im} \sum_{j=0}^{\Im} [\mathbf{r}_i(j) - \mathbf{r}_i]^2 \right]^{1/2} \tag{5}$$

The RMSF of the system was performed to understand the displacement and stability of each protein residue in the trajectory of the 100 ns simulation.

Figure 7 displays the primary interactions of the major coffee ground oil composition complexes studied, demonstrating significant conformational changes of the compound–Eversa lipase complexes during the simulation. The simulation trajectories of all complexes showed mean oscillations with substantial correlations with critical replication residues. The only complexes with values higher than 2.0 Å for residue His 268 were those formed between methyl decanoate and Eversa. Additionally, lipase complexes with methyl docosanoate and 6,9-methyl octadienoate showed RMSF values above 2.5 Å in residues Phe 63, Thr 102, and Asn 232, despite the fluctuations observed. Nonetheless, the results suggested that the structures remained stable in an aqueous solution. When complexed with various ligands through docking techniques, the protein conformations obtained from the MD simulations provided crucial information about the small molecules' binding modes in different enzyme folding states [63,64].

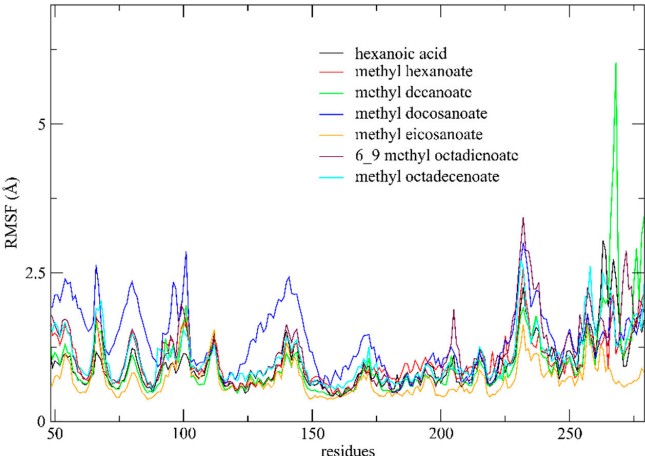

**Figure 7.** Root Mean Square Fluctuation (RMSF) concerning the initial confirmation of the ligand–enzyme complex versus the simulation time (ns) in the production simulations step of the MD with coffee ground oil composition/Eversa lipase.

### 3.4.3. H-bonds Analysis

The number of hydrogen bonds is essential to verify whether a complex has reached stability in a dynamic system [63,64]. In the Figure 8A–G are shown the graphs of hydrogen bonds in relation to the time of 100 ns with their moving averages.

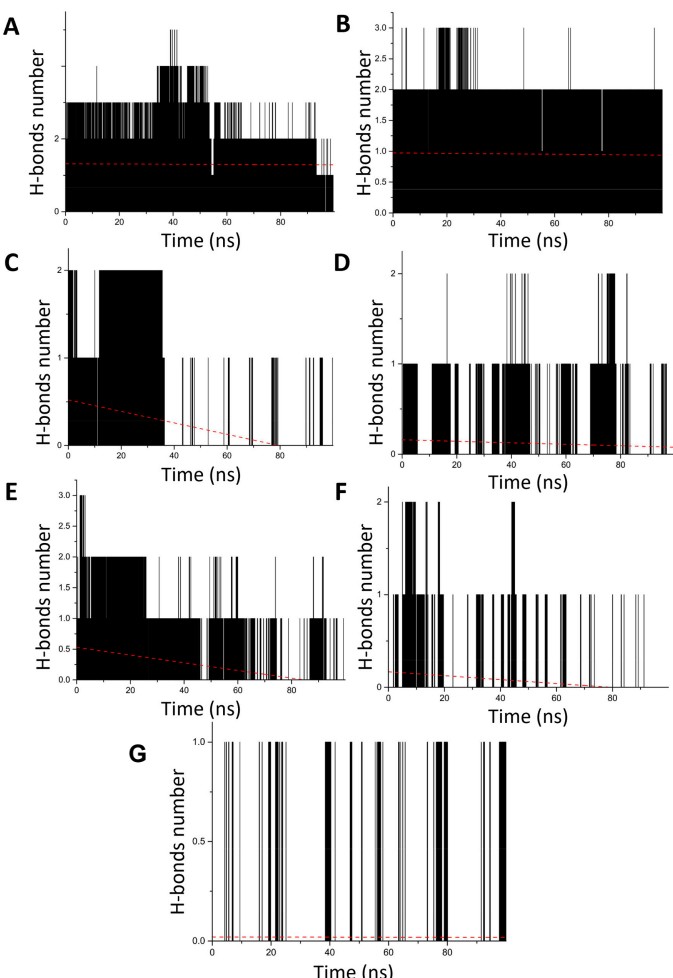

**Figure 8.** Hydrogen bonds formed between the protein and the ligand during the simulation steps. (**A**) hexadecanoic acid; (**B**) methyl decanoate; (**C**) methyl eicosanoate; (**D**) methyl hexadecanoate; (**E**) 6,9-methyl octadienoate; (**F**) methyl octadecanoate; (**G**) methyl docosanoate.

After the 100 ns production stages, the formation of hydrogen bonds between Eversa lipase and the corresponding simulated ligands was observed in Figure 8 to verify stability in the dynamic system. The hydrogen-bonding networks changed during the simulation, and the number of interactions fluctuated between three and five for the lipase. The Eversa with hexanoic acid complex (Figure 8A) displayed isolated hydrogen bonds and a moderate average number of hydrogen bonds per period (up to five), indicating relatively adequate and median hydrogen-bonding networks forming reasonable connections during its trajectory (red line). The MD simulations of methyl decanoate and 6,9-methyl octadienoate (Figure 8E) showed more interactions along the course (three links), suggesting a hydrogen-bonding network moderately more significant than the previous one.

The simulation of other acids with the Eversa enzyme resulted in a weaker network of interactions, with a maximum of two hydrogen bonds formed. However, the stability of the complex was maintained due to the presence of these connections, even though the size and functionalities of the compounds were not as significant [65,66].

Therefore, complementary correlations can be observed when comparing hydrogen bonds formed in molecular dynamics with those previously obtained by the coupling process, indicating the convergence of a static method to a continuous system process.

### 3.4.4. SASA Calculations

Solvent-Accessible Surface Area (SASA) measures the surface area of a protein that is accessible to solvent molecules and plays a critical role in understanding the behavior of proteins in solution. The SASA of the coffee bean oil composition complexes was monitored during 100 ns of molecular dynamics (MD) simulations, a powerful computational technique that can provide detailed insights into the dynamics and stability of biomolecules [67,68].

The results of the SASA analysis demonstrated unique patterns (Figure 9). Notably, hexadecanoic acid, methyl octadecanoate, methyl docosanoate, 6,9-methyl octadienoate, and methyl octadecenoate exhibited a significant rise in SASA values during the simulation, indicating relaxation of the structure. Conversely, the SASA values for methyl eicosanoate and methyl hexadecanoate decreased, indicating tension in the enzyme upon complex formation [69,70].

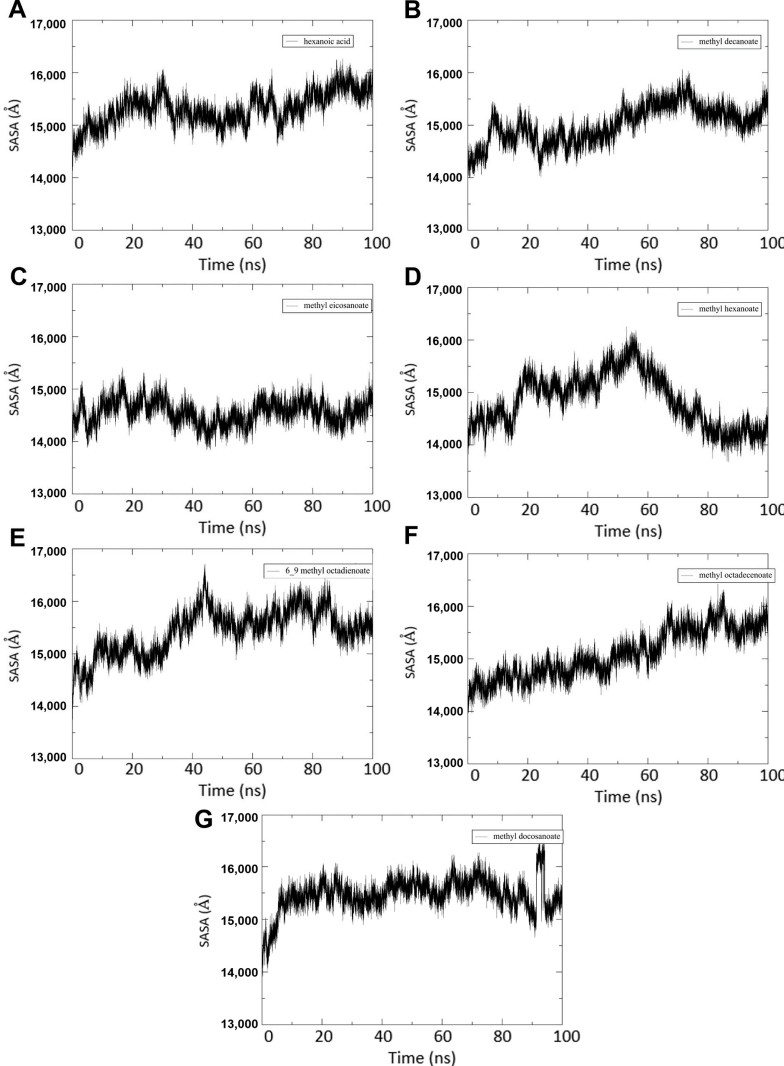

**Figure 9.** Solvent-Accessible Surface Area (SASA) of the Eversa lipase as a function of time from the MD simulations. The curves are running raw data averages with a window of 100 ns. (**A**) hexadecanoic acid; (**B**) methyl decanoate; (**C**) methyl eicosanoate; (**D**) methyl hexadecanoate; (**E**) 6,9-methyl octadienoate; (**F**) methyl octadecanoate; (**G**) methyl docosanoate.

Interestingly, the binding of the ligands did not lead to a significant change in the SASA values. This suggests that the binding of the ligands to the protein did not significantly alter the accessibility of the solvent molecules to the protein surface [71,72].

After 100 ns of simulation, the SASA values fluctuated around a steady state value, indicating that the simulated systems were in equilibrium. However, it is essential to note that Eversa molecules with stabilizing monovalent ions had the highest SASA values. The systems with higher ion concentrations had smaller areas, which could have caused the protein structures to shrink under the influence of the surface charge, resulting in more compact structures.

Upon further analysis of the data, it was observed that the fluctuation or expansion of the relaxed surface area was predominantly caused by the fluctuation of the SASA in the flexible C-terminal region. This suggests that the flexibility of the C-terminal region is a critical factor in determining the overall SASA values of the protein.

These findings could potentially guide the development of new compounds that interact more efficiently with the protein surface or help optimize the conditions for complex formation.

In summary, the analysis of Solvent-Accessible Surface Area (SASA) is highly relevant in molecular dynamics simulations as it provides valuable information about biomolecular interactions, protein folding, ligand binding, protein–protein interactions, and solvent effects. It helps in understanding the structure–function relationships of biomolecules and aids in various areas of molecular biology and drug design.

### 3.4.5. MM/GBSA Calculations

One way to determine the free energies of a receptor complex is by using the MM/GBSA method, which involves calculating molecular mechanics energies along with generalized Born and surface area continuum solvation. The software tool MolAICal uses this approach to rapidly estimate the free energy of a system using three trajectories obtained from molecular dynamics simulations without considering ligand entropy [69,70].

Based on free energy calculations, the complex formed between Eversa and methyl octadecenoate showed the best result with a value of −26.86 kcal/mol. The methyl eicosanoate/Eversa and 6,9-methyl octadienoate/Eversa complexes had lower free energies of −23.62 kcal/mol and −23.41 kcal/mol, respectively. The other complexes formed in the simulations had free energies ranging from −16.26 to −13.27 kcal/mol, as shown in Table 3.

**Table 3.** Oil composition and molecular docking results.

| Complex | $\Delta E_{ele} + \Delta G_{sol}$ (kcal/mol) | $\Delta E_{vdw}$ (kcal/mol) | $\Delta G_{bind}$ (kcal/mol) | Standard Deviation |
|---|---|---|---|---|
| hexadecanoic acid/Eversa | 18.43 | −29.14 | −13.27 | +/− 0.052 |
| methyl hexadecanoate/Eversa | 17.67 | −33.93 | −16.26 | +/− 0.023 |
| methyl octadecanoate/Eversa | 10.33 | −37.19 | −26.86 | +/− 0.027 |
| methyl docosanoate/Eversa | 20.76 | −39.79 | −19.03 | +/− 0.024 |
| methyl eicosanoate/Eversa | 13.75 | −37.37 | −23.62 | +/− 0.027 |
| 6,9-methyl octadienoate/Eversa | 11.97 | −35.38 | −23.41 | +/− 0.026 |
| methyl octadecenoate/Eversa | 19.56 | −35.03 | −15.47 | +/− 0.026 |

It is important to note that in this analysis of the entire trajectory, the ligands that improved affinity with the enzyme Eversa were different from the molecular docking study, except 6,9-methyl octadienoate. Figure 10 shows the interactions of the complexes that

stood out the most with a record in the trajectory between 1 ns, 50 ns, and 100 ns: methyl octadienoate/Eversa, methyl eicosanoate/Eversa, and 6,9-methyl octadienoate/Eversa.

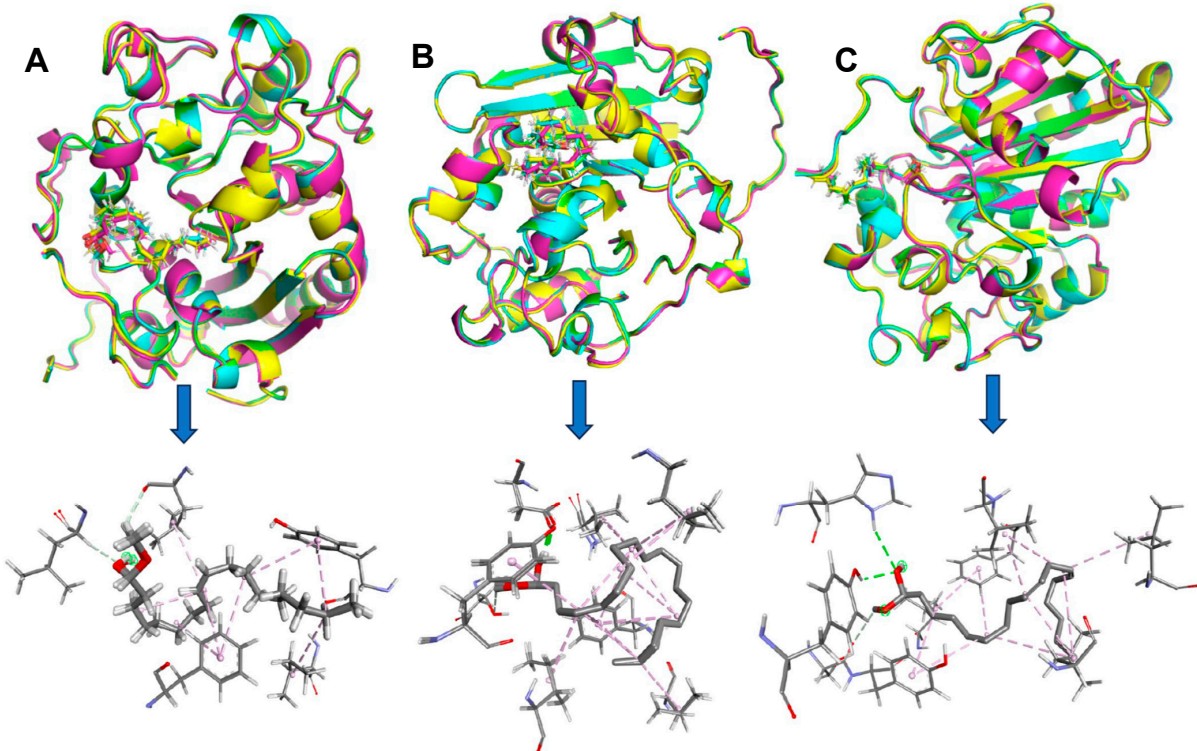

**Figure 10.** Interactions between methyl octadecenoate: (**A**) methyl eicosanoate; (**B**) 6,9-methyl-octadienoate; (**C**) Eversa amino acid residues between 1–10 ns of trajectory.

The phrase "entropy variation" pertains to the decrease in available degrees of freedom due to the establishment of one or more interactions. Before the formation of a complex, only two molecules (ligand and protein) could move in various ways, including rotation, translation, and vibration. However, after the formation of a complex, the movement of the molecules is restricted.

This estimate can be obtained from the typical mode calculations for the two systems. Thus, for a macromolecular complex with a target and a ligand, the interaction energy must be estimated according to Equations (6)–(8) [71,72].

$$\Delta A_{interaction}^{(vac)} = \left( E_{complex}^{MM} - E_{target}^{MM} \right) - \left( E_{complex}^{MM} - E_{ligand}^{MM} \right) + T\Delta S_{NORMODS} \tag{6}$$

$$\Delta A_{interaction}^{(vac)} = E_{complex}^{MM} - E_{target}^{MM} - E_{complex}^{MM} + E_{ligand}^{MM} + T\Delta S_{NORMODS} \tag{7}$$

$$\Delta A_{interaction}^{(vac)} = E_{ligand}^{MM} - E_{target}^{MM} + T\Delta S_{NORMODS} \tag{8}$$

## 4. Conclusions

This study effectively modeled the interactions between the oils present in coffee grounds and Eversa® Transform 2.0, showing an affinity of each oil with the active site of the enzyme and proving to be important information for future research on reaction catalysis for producing biodiesel or biolubricants. The study found that 6,9-methyl octadienoate binds near the enzyme's active site with favorable free energy and specific interactions. Molecular dynamics simulations showed stability and low RMSD values, indicating suitable coupling positions for the reaction. According to the MMGBSA study, methyl octadecenoate was the best free energy estimate for the enzyme–substrate complex. The CASTp analysis

identified the main active sites for potential immobilization of the enzyme in experimental studies, excluding the native region of the ligand. However, it is essential to note that while computer simulations provide useful initial screening tools, in vitro and extended applications may face additional variables and challenges that require further study and consideration, such as transport conditions, enzymatic inhibition, equilibrium conditions, and physicochemical properties of the produced biodiesel.

**Author Contributions:** Conceptualization, M.M.R.N.; methodology, A.F.d.S., A.M.M. and F.L.B.d.S.; software, A.M.d.F. and E.S.M.; validation, R.P.C., A.M.d.F., I.M.L. and E.S.M.; formal analysis, M.C.M.d.S. and J.C.S.d.S.; investigation, J.C.S.d.S. and R.L.F.M.; resources, M.M.R.N.; data curation, J.C.S.d.S.; writing—original draft preparation, R.L.F.M.; writing—review and editing, J.C.S.d.S. and R.L.F.M.; visualization, J.C.S.d.S. and A.M.d.F.; supervision, J.C.S.d.S., A.M.d.F. and M.C.M.d.S.; project administration, A.M.d.F.; funding acquisition, J.C.S.d.S. and A.M.d.F. All authors have read and agreed to the published version of the manuscript.

**Funding:** This research received no external funding.

**Institutional Review Board Statement:** Not applicable.

**Informed Consent Statement:** Not applicable.

**Data Availability Statement:** Not applicable.

**Acknowledgments:** The authors would like to express their gratitude for support that was provided through the Instituto de Engenharias e Desenvolvimento Sustentável (IEDS) at the Universidade da Integração Internacional da Lusofonia Afro-Brasileira (UNILAB). We gratefully acknowledge the following Brazilian Agencies for Scientific and Technological Development: Fundação Cearense de Apoio ao Desenvolvimento Científico e Tecnológico (FUNCAP) (PS1-0186-00216.01.00/21), Conselho Nacional de Desenvolvimento Científico e Tecnológico (CNPq) (307454/2022-3), and Coordenação de Aperfeiçoamento de Ensino Superior (CAPES) (finance code 001).

**Conflicts of Interest:** The authors declare no conflict of interest.

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
