# Peer review of "Ester Production Using the Lipid Composition of Coffee Ground Oil (Coffea arabica): A Theoretical Study of Eversa® Transform 2.0 Lipase as an Enzymatic Biocatalyst"

_compounds, doi:10.3390/compounds3030031_

Round 1
Reviewer 1 Report (Previous Reviewer 4)
The paper has been corrected and updated with regards to previous submission, however the presentation of the results still needs to be improved including: the necessary thorough interpretation of the results (eg. MD part - the ligands appear to be quite unstable); figures/graphs quality could be higher so as to increase visibility and the their understanding; language and minor editorial changes are necessary - eg in. Figures 4, 5 and 10 (that show the calculated/putative binding poses and/or interactions) the description should be adequately adjusted.
Lastly, but most important - it appears that the obtained model, apart of stereochemical and structural evaluations has not been validated by means of the ability to distinguish active from inactive compounds.
The language can be improved
Author Response
Answer: Dear reviewers, we adjusted the quality of the requested figures. Regarding stereochemistry, in 2D images, the Discovery Studio program does not differentiate Cis from Trans, but in 3D projections, all are Cis, which represents the interactions of each molecule.

Reviewer 2 Report (Previous Reviewer 3)
This manuscript is an improvement of the original version, but there are still some issues that need to be addressed.
1) Line 146-152: this section describes the ANOVA analysis, but the ANOVA analysis was not performed in this work.
2) Line 306-308: the computational details should be moved to the method sections.
3) Figure 8. The authors made a separate panel for each plot, however it is still not clear, especially panels A and B. I believe it's better to plot the moving average, or even just box plot for the first and second half of the simulations.
4) Figure 9. Only one simulation was performed for each ligand, so the SASA patterns may not be reproducible. In addition, the authors did not explain how the SASA patterns correlate with the ligand binding affinity, or how they could be used to guide the developments of new compounds.
Author Response
Comments and Suggestions for Authors
This manuscript is an improvement of the original version, but there are still some issues that need to be addressed.
1) Line 146-152: this section describes the ANOVA analysis, but the ANOVA analysis was not performed in this work.
Answer: Dear all, thank you for the observation. Therefore, we removed this item because it was a mistake..
2) Line 306-308: the computational details should be moved to the method sections.
Answer: Thanks for the observation, we have already made the appropriate change as requested.
3) Figure 8. The authors made a separate panel for each plot, however it is still not clear, especially panels A and B. I believe it's better to plot the moving average, or even just box plot for the first and second half of the simulations.
Answer: Dear reviewers, the mean RMSD value was added for each panel.
4) Figure 9. Only one simulation was performed for each ligand, so the SASA patterns may not be reproducible. In addition, the authors did not explain how the SASA patterns correlate with the ligand binding affinity, or how they could be used to guide the developments of new compounds.
Answer: Our study of Solvent-accessible surface area (SASA) fluctuations per molecule for 100ns showed the hydrophobic interaction with solvents, ligands, and enzymes and how the conformations behaved throughout the trajectory to relax or compress. Therefore, additional studies, such as density, energy, and volume, should be done.

Reviewer 3 Report (Previous Reviewer 2)
The manuscript has significantly improved.
It is now ready for publication.
Best wishes,
Minor editing of English language required
Author Response
Answer: Thank you for your contributions and comments. Thank you for noting the relevance of the article and accepting it.

Reviewer 4 Report (New Reviewer)
This study modeled the interactions between the oils present in coffee grounds and Eversa® Transform 2.0, showing an affinity of each oil with the active site of the enzyme and proving to be important information for future research on reaction catalysis for producing biodiesel or biolubricants. The authors need to clearly indicate the novelty of their work at the end of the introduction section. There are many issues that should be considered to update this paper:
1. This paper needs an English editing, there are many grammatical mistakes along the paper.
2. The authors have cited 112 references which is quit high number of references for a research article, even though this can be accepted, however the reason of adding each reference should be clearly known. Therefore, please eliminate multiple references. After that, please check the manuscript thoroughly and eliminate the lumps in the manuscript. This should be done by characterizing each reference individually and by mentioning 1 or 2 phrases per reference to show how it is different from the others and why it deserves mentioning. Multiple references are of no use for a reader and can substitute even a kind of plagiarism, as sometimes authors are using them without proper studies of all references used. In the case, each reference should be justified by it is used and at least short assessment provided. Please avoid lump reference (2 or more references together). Explain them individually. Example for this is section 1 such as references (6-9, 25-30,…..etc).
3. In the introduction section, line 37, (Authors: Coffee consumption in Europe in 2020 and 2021 was 54.065 thousand bags), does the consumption in Europe for years 2020 and 2021 is the same? the authors mentioned the consumptions in bags, therefore, what does the bag weigh?
4. Introduction, lime 38, (Authors: representing an annual growth of 0.5% compared to previous years), please specify the years, you have to be specific in comparison.
5. Introduction, line 41, why does the industry have not used the coffee grounds till now? I suggest you discuss the reasons and challenges.
6. Introduction, line 41, (Authors: However, they are rich in fatty acids, such as lignin, cellulose, and hemicellulose). Lignin, cellulose, and hemicellulose are not types of fatty acids, please remove the word “Such as”
7. Introduction, line 50, (Authors: In order to use FFA derived from coffee grounds as a biofuel), please add the word source at the end of the sentence, as FFA is not a fuel by its own, however they need to be converted to biodiesel.
8. Introduction, lines 54,57, (Authors: Although enzymatic catalysts still have a high cost, the scientific community has been working……….), authors mentioned that immobilization is an opportunity to reduce the high cost of the enzymes utilization, please also add other factors that might reduce the overall cost of the enzymatic process such as reactor configuration, reaction engineering,….etc.
9. Introduction, lines 79-82, (Authors: to catalyze 79 seven free fatty acids extracted from coffee grounds, including hexadecanoic acid, methyl hexadecanoate, methyl octadecanoate, methyl docosanoate, methyl eicosanoate, methyl octadienoate, and methyl octadecenoate). The following are not fatty acids as the authors mentioned, however they are fatty esters (methyl hexadecanoate, methyl octadecanoate, methyl docosanoate, methyl eicosanoate, methyl octadienoate, and methyl octadecenoate). Please modify. You can also replace the word fatty acids with fatty matters.
1 I am quite confused about if this research work has been received a fund, because authors declared that they have received no fund by mentioning that (Funding: This research received no external funding), and then they acknowledged the Fundação Cearense de Apoio ao Desenvolvimento Científico e Tecnológico 468 (FUNCAP, PS1-00186-00216.01.00/21), Conselho Nacional de Desenvolvimento Científico e 469 Tecnológico (CNPq, 311062/2019-9), and Coordenação de Aperfeiçoamento de Pessoal de Nível 470 Superior (CAPES-Finance Code 001) in the acknowledgment section.
This paper needs an English editing, there are many grammatical mistakes along the paper.
Author Response
Comments and Suggestions for Authors
This study modeled the interactions between the oils present in coffee grounds and Eversa® Transform 2.0, showing an affinity of each oil with the active site of the enzyme and proving to be important information for future research on reaction catalysis for producing biodiesel or biolubricants. The authors need to clearly indicate the novelty of their work at the end of the introduction section. There are many issues that should be considered to update this paper:
Answer: Thank you for your contributions and comments. Thank you for noting the relevance of the paper.
- This paper needs an English editing, there are many grammatical mistakes along the paper.
Answer: Thank you for your contributions and comments. The paper has been revision in English.
- The authors have cited 112 references which is quit high number of references for a research article, even though this can be accepted, however the reason of adding each reference should be clearly known. Therefore, please eliminate multiple references. After that, please check the manuscript thoroughly and eliminate the lumps in the manuscript. This should be done by characterizing each reference individually and by mentioning 1 or 2 phrases per reference to show how it is different from the others and why it deserves mentioning. Multiple references are of no use for a reader and can substitute even a kind of plagiarism, as sometimes authors are using them without proper studies of all references used. In the case, each reference should be justified by it is used and at least short assessment provided. Please avoid lump reference (2 or more references together). Explain them individually. Example for this is section 1 such as references (6-9, 25-30,…..etc).
Answer: Thank you for your contributions and comments. A strong review of the references was carried out and adjusted.
- In the introduction section, line 37, (Authors: Coffee consumption in Europe in 2020 and 2021 was 54.065 thousand bags), does the consumption in Europe for years 2020 and 2021 is the same? the authors mentioned the consumptions in bags, therefore, what does the bag weigh?
Answer: Thank you for your comments and contributions. The first paragraph has been adjusted with new sentences and updated references.
- Introduction, lime 38, (Authors: representing an annual growth of 0.5% compared to previous years),please specify the years, you have to be specific in comparison.
Answer: Thank you for your comments and contributions. The first paragraph has been adjusted with new sentences and updated references.
- Introduction, line 41, why does the industry have not used the coffee grounds till now? I suggest you discuss the reasons and challenges.
Answer: Thank you for your comments and contributions. The first paragraph has been adjusted with new sentences and updated references.
- Introduction, line 41, (Authors: However, they are rich in fatty acids, such as lignin, cellulose, and hemicellulose). Lignin, cellulose, and hemicellulose are not types of fatty acids, please remove the word “Such as”
Answer: Thank you for your comments and contributions. The first paragraph has been adjusted with new sentences and updated references.
- Introduction, line 50, (Authors: In order to use FFA derived from coffee grounds as a biofuel), please add the word source at the end of the sentence, as FFA is not a fuel by its own, however they need to be converted to biodiesel.
Answer: Thank you for your comments and contributions. The sentence has been adjusted.
- Introduction, lines 54,57, (Authors: Although enzymatic catalysts still have a high cost, the scientific community has been working……….), authors mentioned that immobilization is an opportunity to reduce the high cost of the enzymes utilization, please also add other factors that might reduce the overall cost of the enzymatic process such as reactor configuration, reaction engineering,….etc.
Answer: Thank you for your comments and contributions. The comment has been added and adjusted.
- Introduction, lines 79-82, (Authors: to catalyze 79 seven free fatty acids extracted from coffee grounds, including hexadecanoic acid, methyl hexadecanoate, methyl octadecanoate, methyl docosanoate, methyl eicosanoate, methyl octadienoate, and methyl octadecenoate). The following are not fatty acids as the authors mentioned, however they are fatty esters (methyl hexadecanoate, methyl octadecanoate, methyl docosanoate, methyl eicosanoate, methyl octadienoate, and methyl octadecenoate). Please modify. You can also replace the word fatty acids with fatty matters.
Answer: Thank you for your comments and contributions. The comment has been added and adjusted.
- I am quite confused about if this research work has been received a fund, because authors declared that they have received no fund by mentioning that (Funding: This research received no external funding), and then they acknowledged theFundação Cearense de Apoio ao Desenvolvimento Científico e Tecnológico 468 (FUNCAP, PS1-00186-00216.01.00/21), Conselho Nacional de Desenvolvimento Científico e 469 Tecnológico (CNPq, 311062/2019-9), and Coordenação de Aperfeiçoamento de Pessoal de Nível 470 Superior (CAPES-Finance Code 001) in the acknowledgment section.
Answer: Thank you for your comments and contributions. The sentence has been adjusted.

Round 2
Reviewer 1 Report (Previous Reviewer 4)
Why did the authors use outdated modeling algorigthm instead of eg. AplhaFold or its modifications models?
please spell check the manuscript using one of freely available AI=powered translators
Author Response
Comments and Suggestions for Authors
Why did the authors use outdated modeling algorigthm instead of eg. AplhaFold or its modifications models?
Answer: Thanks for the comment and contribution. Despite being older, the software used presents results that can be validated, as we follow all established parameters. In addition, given our interests, the software presents accuracy in predicting structures, speed of obtaining results and scalability.
Comments on the Quality of English Language
Answer: Thanks for the comment and contribution. The English language quality has been adjusted.

Reviewer 2 Report (Previous Reviewer 3)
The concerns raised in the last round of review have not been fully addressed in this version, therefore significant improvements are needed before the manuscript can be recommended for publication.
(1) The ANOVA analysis section, which is irrelevant to this study, is still in the current version. The authors mentioned in their response letter that this section was removed, so it is likely that an incorrect version was uploaded.
(2) It was suggested that a a plot of moving average is better for Figure 8. However, the authors added average values (which are not the same as moving average) in Figure 6, while Figure 8 remain unchanged.
(3) The authors still do not concretely explain the relevance of the SASA analysis. The authors also mentioned in the response letter that "Therefore, additional studies, such as density, energy, and volume, should be done." However, none of these quantities (density, energy, volume) are useful measures for ligand binding affinity.
Author Response
Reviewer #2
Comments and Suggestions for Authors
The concerns raised in the last round of review have not been fully addressed in this version, therefore significant improvements are needed before the manuscript can be recommended for publication.
(1) The ANOVA analysis section, which is irrelevant to this study, is still in the current version. The authors mentioned in their response letter that this section was removed, so it is likely that an incorrect version was uploaded.
Answer: Thanks for the comment and contribution. We apologize and confirm the withdrawal of all the ANOVA section.
(2) It was suggested that a a plot of moving average is better for Figure 8. However, the authors added average values (which are not the same as moving average) in Figure 6, while Figure 8 remain unchanged.
Answer: Thanks for the comment and contribution. We apologize for not inserting the moving average of figure 8, follow the figures already adapted with the average.
(3) The authors still do not concretely explain the relevance of the SASA analysis. The authors also mentioned in the response letter that "Therefore, additional studies, such as density, energy, and volume, should be done." However, none of these quantities (density, energy, volume) are useful measures for ligand binding affinity.
Answer: Thanks for the comment and contribution. This sentence was added in the article on page 440-445.
“In summary, the analysis of Solvent Accessible Surface Area is highly relevant in molecular dynamics simulations as it provides valuable information about biomolecular interactions, protein folding, ligand binding, protein-protein interactions, and solvent effects. It helps in understanding the structure-function relationships of biomolecules and aids in various areas of molecular biology and drug design.”

Reviewer 4 Report (New Reviewer)
The authors have addressed all the concerns of the reviewer.
Author Response
Reviewer #4
Comments and Suggestions for Authors
The authors have addressed all the concerns of the reviewer.
Answer: Thank you for your contributions and comments. Thank you for noting the relevance of the article and accepting it.

Round 3
Reviewer 2 Report (Previous Reviewer 3)
The authors addressed most of my comments. However, the quality of Figure 8 is still very poor. Please remove the watermark "Speed mode is on", and either remove the table or make the font size in the table larger.
Author Response
Dear Editor
Compounds (ISSN 2673-6918)
We are sending the fourth round (R4) of revision of our manuscript:
Manuscript Number: compounds-2386603
Title: Ester Production Using the Lipid Composition of Coffee Ground Oil (Coffea arabica): A theoretical study of Eversa Transform 2.0 lipase as na enzymatic biocatalyst
Reviewer #2
Comments and Suggestions for Authors
The authors addressed most of my comments. However, the quality of Figure 8 is still very poor. Please remove the watermark "Speed mode is on", and either remove the table or make the font size in the table larger.
Answer: Thank you for your contributions and comments. We apologize for the error in figure 8. We have resolved the issue and improved the quality.

This manuscript is a resubmission of an earlier submission. The following is a list of the peer review reports and author responses from that submission.
Round 1
Reviewer 1 Report
The study have a merit as a preliminary investigation only, it doesn’t not meet standards of significance, no real crystal structure for the enzyme isolated till the moment, limited number of the organic compounds, no evidence for future applicability of the obtained results
Needs polishing.
Author Response
Reviewer #1:
The study have a merit as a preliminary investigation only, it doesn’t not meet standards of significance, no real crystal structure for the enzyme isolated till the moment, limited number of the organic compounds, no evidence for future applicability of the obtained results.
Answer: Dear reviewer, we understand your request. However, the enzyme Eversa® Transform 2.0 lipase lipase does not yet have its crystallography, as it is a new enzyme. Making the paper have a good relevance. We added new references in the text that support the paper in question. The paper is an initial work that can support new research using Eversa.
Comments on the Quality of English Language
Answer: Thank you for your contributions and comments. The paper has been revised in English.

Reviewer 2 Report
Dear authors,
Your manuscript needs extensive editing in the English language. In addition to this, the methods are not adequately described, and much work needs to be done there. Some words and phrases need to be in italics. Please move Figures 6,8,9 to the supplementary section.
Best wishes,
Your manuscript needs extensive editing in the English language.
Author Response
Reviewer #2:
Some words and phrases need to be in italics.
Answer: Thank you for your comments and contributions. Your suggestion has been added to the text.
Your manuscript needs extensive editing in the English language.
Answer: Thank you for your contributions and comments. The paper has been revision in English.

Reviewer 3 Report
This manuscript presents a molecular docking and molecular dynamics simulations study for the binding between several lipids in coffee beans and Eversa lipase. While the aim of the study was to investigate the potential use of Eversa lipase for biofuel production using coffee ground oil, the actual study design is only concerned with the ranking of several components in coffee ground oil in terms of binding affinity, and which has no direct link to the stated aim. Predicting the affinity between the free fatty acids (FFA) and Eversa Transform 2.0 is not particularly interesting. Eversa Transform 2.0 is a well-studied commercial product designed to be efficient for FFAs, and there is no apparent reason why it cannot be applied to the FFAs in coffee ground oil. And as the authors stated, the main approaches for optimizing the biocatalyst has been immobilizing the enzymes, which is not related to the binding affinity. Nevertheless, it might be interesting to study the catalytic mechanism to help design better enzymes, which would require QM or QM/MM calculations of the transition state. Therefore, I encourage the authors to rethink about the research questions and experiment design.
Some minor points/questions
(1) Line 149-155: The statistical analysis for animal experiments seems unrelated to the current study.
(2) Figure 8 (Line 367-368) is unreadable because the lines are not clearly separated. Please consider plotting moving averages.
(3) Was the SASA calculation based on protein only or protein-ligand complex? If it was on protein only, then it does not tell the stability of the complex. As the authors stated, the change in SASA was due to "the flexibility of the C-terminal region" (Line 393), which might not affect the binding.
(4) The interactions between the ligand and protein were analyzed based on docking structures. Since the structures in MD simulations were different, and there seems to be a gradual loss of hydrogen bonds (Figure 8), the same analysis should also be done on MD structures.
The language is fine.
Author Response
Reviewer #3:
This manuscript presents a molecular docking and molecular dynamics simulations study for the binding between several lipids in coffee beans and Eversa lipase. While the aim of the study was to investigate the potential use of Eversa lipase for biofuel production using coffee ground oil, the actual study design is only concerned with the ranking of several components in coffee ground oil in terms of binding affinity, and which has no direct link to the stated aim.
Answer: Thank you for your contributions and comments. The paper had a formulation in the summary and conclusions to better identify the objective of the text. Understanding your points, however, the paper aimed to be an initial research of Eversa lipase (which is a new commercial lipase) and of Coffea arabica that has not yet been studied along with this lipase.
Nevertheless, it might be interesting to study the catalytic mechanism to help design better enzymes, which would require QM or QM/MM calculations of the transition state. Therefore, I encourage the authors to rethink about the research questions and experiment design.
Answer: Thank you for your contributions and comments. We understand your points, however the paper was intended to be an initial study. Eversa is a new lipase and studies on it are still beginning. Some papers were added in the introduction to support the text in question.
Some minor points/questions
(1) Line 149-155: The statistical analysis for animal experiments seems unrelated to the current study.
Answer: Thank you for your comments and contributions. Your suggestion has been added to the text.
(2) Figure 8 (Line 367-368) is unreadable because the lines are not clearly separated. Please consider plotting moving averages.
Answer: Thank you for your comments and contributions. The figures were posted individually for better understanding. And now they have been changed to Figure 8a, 8b, 8c, 8c, 8d, 8e, 8f and 8g.
(3) Was the SASA calculation based on protein only or protein-ligand complex? If it was on protein only, then it does not tell the stability of the complex. As the authors stated, the change in SASA was due to "the flexibility of the C-terminal region" (Line 393), which might not affect the binding.
Answer: Thank you for your comments and contributions. The sentence has been corrected in the manuscript.
(4) The interactions between the ligand and protein were analyzed based on docking structures. Since the structures in MD simulations were different, and there seems to be a gradual loss of hydrogen bonds (Figure 8), the same analysis should also be done on MD structures.
Answer: Thank you for your comments and contributions. New comments on this observation were added, in addition to the insertion of Figure 10a, 10b, and 10c.

Reviewer 4 Report
Presented herein study to determine possible interactions of coffee grounds FFA’s with Eversa® Transform 2.0 enzyme, is characterised by a well-defined scientific objective and the conduct of research. It is noteworthy that the authors used freely available software in its entirety. However, I have a few comments and questions.
First of all – how was the model obtained using MODELLER validated? Moreover, what drove the authors to choose a slightly outdated algorithm, when many more precise methods or even ready-to-use computated structure models libraries are available?
Some minor corrections would include:
Chapter 2.1.1. Latin species names should be italic through the manuscript
Line 132 – please rephrase “The software was used…” – unclear what software
Line 163 – BLA ligand – please explain or give chemical name
Line 208 – “These results support the reliability of the protein model obtained.” I don’t agree with that – ~91% AA in favoured regions is considered too low by some authors. At least enrichment test would be accurate to estimate the quality and usability of protein .
Line 223 – what is DS software mentioned here?
Figure 5 – please use the same projection of both ligands in the binding site, as the ones are presented reverted in x and y axis as well.
Figures 6 & 9 - Please break down the chart into individual ligands. In this form it is unreadable. Yet from what I can see the ligands were not that stable during the simulations.
An additional Figure (for each of the compounds, might be in Supplementary Materials) depicting superimposition of ligands in individual snapshots would be highly applicable.
Author Response
Reviewer #4:
Comments and Suggestions for Authors
Presented herein study to determine possible interactions of coffee grounds FFA’s with Eversa® Transform 2.0 enzyme, is characterised by a well-defined scientific objective and the conduct of research. It is noteworthy that the authors used freely available software in its entirety. However, I have a few comments and questions.
Answer: Thank you for your contributions and comments. Thank you for noting the relevance of the paper.
First of all – how was the model obtained using MODELLER validated? Moreover, what drove the authors to choose a slightly outdated algorithm, when many more precise methods or even ready-to-use computated structure models libraries are available?
Answer: Homology modeling is based on the concept of molecular evolution. That is, it is assumed that the similarity between the primary structures of this protein and homologous proteins of known three-dimensional structures (template proteins) implies structural similarity between them. For Homology Modeling, we used 4 processing steps until its validation, based on the literature, as shown in the methodology in the items: 2.1.1. Identification and selection of protein-fold, 2.1.2. Alignment of Target and Mold Sequences, 2.1.3. Model Construction and Optimization and 2.1.4. Protein Validation, performed by PROCHECK.
Some minor corrections would include:
Chapter 2.1.1. Latin species names should be italic through the manuscript
Answer: Thank you for your comments and contributions. The sentence has been corrected in the manuscript.
Line 132 – please rephrase “The software was used…” – unclear what software
Answer: Thank you for your comments and contributions. The sentence has been corrected in the manuscript.
Line 163 – BLA ligand – please explain or give chemical name
Answer: The sentence was withdrawn, as it is a mistake. Thanks for the correction.
Line 208 – “These results support the reliability of the protein model obtained.” I don’t agree with that – ~91% AA in favoured regions is considered too low by some authors. At least enrichment test would be accurate to estimate the quality and usability of protein.
Answer: Thank you for your comments and contributions. Although the value is considered poor by some authors, the degree of identity between the primary structures of proteins-mold and the problem protein is equal to or greater than about 25%, when the number of residues is greater than 80, there is a large probability that these proteins have structures similar three-dimensional. In addition, it was visualized the obtaining of a well-refined structure. The G-factor indicates that the stereochemical properties obtained were considered normal.
Line 223 – what is DS software mentioned here?
Answer: Thank you for your comments and contributions. The sentence has been corrected in the manuscript.
Figure 5 – please use the same projection of both ligands in the binding site, as the ones are presented reverted in x and y axis as well.
Answer: Thank you for your comments and contributions. The graphs representing the molecular docking were arranged in the same projection, according to your suggestion.
Figures 6 & 9 - Please break down the chart into individual ligands. In this form it is unreadable. Yet from what I can see the ligands were not that stable during the simulations.
Answer: Thank you for your comments and contributions. Figures 6 and 9 were placed individually for a better understanding of the work. In this way the figures were created: 6a, 6b, 6c, 6d, 6e, 6f, and 6g, as well as the figures: 9a, 9b, 9c, 9d, 9e, 9f, and 9g.